# A Method to Improve Mounting Tolerance of Open-Type Optical Linear Encoder

**DOI:** 10.3390/s23041987

**Published:** 2023-02-10

**Authors:** Xinji Lu, Artūras Kilikevičius, Fan Yang, Donatas Gurauskis

**Affiliations:** 1Institute of Mechanical Science, Vilnius Gediminas Technical University, LT-03224 Vilnius, Lithuania; 2Changchun Institute of Optics, Fine Mechanics and Physics, Chinese Academy of Sciences, Changchun 130033, China; 3University of Chinese Academy of Sciences, Beijing 100049, China

**Keywords:** open-type optical linear encoder, scanning reticle, gratings

## Abstract

Accuracy becomes progressively important in the wake of development in advanced industrial equipment. A key position sensor to such a quest is the optical linear encoder. Occasionally, inappropriate mounting can cause errors greater than the accuracy grade of the optical linear encoder itself, especially for open-type optical linear encoders, where the mounting distance between the reading head and main scale must be accurately controlled. This paper analyzes the diffraction fields of a traditional scanning reticle made by amplitude grating and a newly designed combined grating; the latter shows a more stable phase in mathematical calculation and simulations. The proposed combined gratings are fabricated in a laboratory and assembled into the reading heads. The experimental results indicate that the mounting tolerance between the reading head and the main scale of the optical linear encoder can be improved.

## 1. Introduction

The finest approach for sensors to measure displacement with high accuracy is possibly through optical technology [1]. Back in the 1950s, optical linear encoders began to be adopted in machine tools for direct position feedback [2]. Today, optical encoders are widely used in numerous applications where position measurements are vital to controls, from printers in households [3] to industrial robot arms [4], high-end 3D printers [5], accelerometers [6], high-end medical instruments, such as MRI [7], and wafer scanners in the semiconductor industry [8]. The resolutions of optical linear encoders should be at least 10 times finer than the accuracy of the designed machines so that control accuracy can be ensured [9]. Consequently, the resolution and accuracy of optical linear encoders have a strong demand for improvement to keep pace with the development of advanced equipment. Over the last decades, the resolution and accuracy of optical linear encoders have been enhanced significantly owing to innovations in electronics and optical technologies. Nevertheless, these achievements can be easily offset by external or unintentional man-made errors, such as temperature variations [10], mechanical vibrations [11], surface pollution [12], imperfect mounting, etc. The errors caused by linear encoders not only reduce the accuracy of equipment but also put restrictions on the operation speed or increase the amount of computational work for machines [13]. In worst-case scenarios, incorrect displacement feedback from optical linear encoders could lead to a decrease in the stability of the machine and have a negative impact on the dynamic controls [14]. Multitudinous research and experiments have been performed to diminish these errors. Some works designed special tools or techniques to keep the linear encoder apart from the source of vibration [15], while others focused on developing algorithms to compensate for the errors induced by vibration [11], temperature variations [16], or both [17]. However, reducing the impact of mounting errors can be a more complicated task for optical linear encoders.

From the perspective of working principles, the smaller the mounting distance between the reading head and main scale, the higher the optical signal amplitudes, and the easier it is for subsequent electronics to process analog signals. Nonetheless, such a requirement is only applicable with an enclosed-type optical linear encoder, wherein the reading head and main scale are connected by mechanical structures, as shown in Figure 1a. On the other hand, the reading head and main scale are independent parts in open-type linear encoder systems and are installed separately on different parts of the equipment with relative motions. With the rich variety in high accuracy machines and multitudinous mounting conditions for open-type optical linear encoders, ensuring all mountings are in perfect positions in all scenarios would be unrealistic. Under some circumstances where the mounting gap between the reading head and main scale is not within the ideal range, the errors can be multiple times greater than the accuracy of optical linear encoders [18] and make other aspects of work to improve the accuracy pointless. Therefore, increasing the mounting gap and tolerance is an important and imminent task.

In this paper, a scanning reticle made by the combination of amplitude and phase grating is proposed. Such an approach can significantly increase the mounting tolerance between the reading head and the main scale. In Section 2, the proposed method is compared with the traditional scanning reticle by theoretical calculations and simulations. The fabrication technology of the proposed scanning reticle is also introduced in this section. Section 3 demonstrates the results from the experiments. The conclusion is given in Section 4.

## 2. Methods

### 2.1. Theoretical Investigation

The foremost economical method to increase the mounting tolerance is perhaps by means of modifying the optical path because it is the only approach that does not add additional parts to the reading head or increase the complexity during the manufacturing process. As illustrated in Figure 2, the LED beam collimated by lens incidence through the scanning reticle travels to the main scale and then reflects to the photocells. The LED, collimating lens, scanning reticle, and photocells are all integrated inside the reading head, as shown in Figure 2. Because the scanning reticle is at the bottom of the reading head, the mounting gap between the reading head and the main scale can be simplified as the distance between the scanning reticle and the main scale. Stable light distribution after passing through the scanning reticle is a prerequisite for generating the Moire fringe, which is the fundamental working principle of the optical linear encoder. That is to say, the light signal should have a stable phase, period, and good contrast when the mounting distance varies.

Both the main scale and scanning reticle have engraved grating lines on the surface so that the Moire fringe can appear when there is a relative motion between the two. The scanning reticle is commonly made by a glass substrate with chrome-plated grating lines (opaque for the incident light) on top. The most used type of scanning reticle by encoder manufacturers is made by amplitude grating, especially in enclosed-type optical linear encoders. As illustrated in Figure 3a, the black stripe in the picture represents the opaque zones, which are typically made of a chrome plate. The gray areas represent the euphotic region, which are the uncovered glass substrates.

The amplitude grating has simplicity in structure and is convenient to fabricate. However, according to the theory of the Talbot effect [19], the image of the scanning reticle can only remain stable at a particular distance. In Equation (1), z represents the Talbot distance, d denotes the pitch period of the grating, λ is the wavelength of the LED, and n is an integer. When n = 1, d and λ are given, then the value of z is equal to one Talbot distance.
(1)$$z=\frac{{2\mathrm{nd}}^{2}}{\lambda }$$

According to the principle of the Talbot effect, the image of grating lines on the scanning reticle will reappear at one Talbot distance or an integral multiple of the Talbot distance. In practical applications for optical linear encoders, as the intensity of light decreases with space, the distance between the scanning reticle and main grating is usually kept within one Talbot distance. Ideally, if the distribution of light can stay unchanged during the entire Talbot distance, the optical linear encoder can have a mounting tolerance of one Talbot distance, which is convenient for installment and can prevent errors caused by incorrect mounting distance. Nonetheless, the mounting tolerance within such a distance is limited. A study by LightTrans revealed [20] the relationship between the distribution of light and the variation in the distance z from Equation (1) within the distance of one Talbot distance. As shown in Figure 4, while the distance of z changes, the distribution of light intensity does not remain stable during the entire Talbot distance. Only in a scenario when z is at the position of one Talbot distance, the image of the grating lines can reappear, and at a distance of z/2, the image pattern is reversed, which means the phase is shifted compared with the original image.

The phase variation in light distribution would be unacceptable for optical linear encoders. The electrical output signals and method to calculate the position values of the optical linear encoder can be described by Equations (2) and (3), where $S_{s}\;$and $S_{c}$ are the electrical output signals, $V_{1}$ and $V_{2}$ are the amplitudes, and x is the measured positional value. The phase variation in the light signal induces a phase shift on electrical output signals. According to these equations, this will ultimately lead to position errors. Furthermore, the variation in the phase and intensity of light can make the optical signal hardly readable for photocells. Consequently, optical linear encoders cannot operate properly with the traditional scanning reticle while mounting gaps vary within the distance of z based on the Talbot theory.
(2)$$\{\begin{array}{c} S_{s}{=V}_{1}\sin(\frac{2\pi x}{d})\\ S_{c}{=V}_{2}\cos(\frac{2\pi x}{d}) \end{array}$$
(3)$$\;x=\frac{d}{2\pi }\tan^{-1}\frac{S_{S}}{S_{c}}$$

A feasible solution is to add additional structures on the scanning reticle to increase the effective working distance. Kao designed a triple-grating optical system, wherein the scanning reticle contains both amplitude and phase gratings, and it has a delicate pattern to ameliorate the unstable distribution of light intensity [21]. Nevertheless, such a pattern can be too sophisticated for manufacturing and time consuming for quality inspection processes after mass production. Therefore, a simplified version of the scanning reticle, which is more convenient for fabrication, is worthy of investigation.

The distribution of light intensity through the scanning reticle can be analyzed in accordance with the scalar diffraction theory. In Figure 3, each white column in both scanning reticles can be considered an independent light source. The phase grating, too, can be approximate to light sources but with a phase difference compared with the previous counterparts. The complex amplitude distribution at the main scale (view plane) can be calculated by a summation from all light sources.

According to Fresnel diffraction, each independent light source on the scanning reticle can be described as:(4)$$\tilde{E}(x,y)=\frac{e^{\mathrm{ikz}}}{i\lambda z}\iint_{0}^{a}\tilde{E}\left(x_{1}{,y}_{1}\right)\exp\left\{\frac{\mathrm{ik}}{2z}\left[{\left({x-x}_{1}\right)}^{2}+{\left({y-y}_{1}\right)}^{2}\right]\right\}{\mathrm{dx}}_{1}{\mathrm{dy}}_{1}$$
where ($x_{1}$,$y_{1}$) represent the coordinate on the scanning reticle, and (x,y) denote the coordinate on the main scale. The optical linear encoder is a single-dimensional displacement sensor. Assuming the measurement direction is along the X-axis, then the distribution of light intensity on the Y-axis can be neglected. As the light source of the optical linear encoder is usually collimated and light intensity can be considered as uniform, $\tilde{E}\left(x_{1}{,y}_{1}\right)$ does not affect the contrast of light distribution on the main scale. Hence, Equation (4) can be rewritten as:(5)$$\tilde{E}\left(x\right)=\frac{e^{\mathrm{ikz}}}{i\lambda z}\int_{0}^{a}\exp\frac{\mathrm{ik}}{2z}{\left({x-x}_{1}\right)}^{2}{\mathrm{dx}}_{1}$$

Since $\frac{e^{\mathrm{ikz}}}{i\lambda z}$ has no impact on the contrast of intensity distribution, Equation (5) can be further simplified as:(6)$$\begin{array}{ll} \tilde{E}{\left(x\right)}^{\prime } & =\int_{0}^{a}\exp\frac{\mathrm{ik}}{2z}{\left({x-x}_{1}\right)}^{2}{\mathrm{dx}}_{1}\\ & =\int_{0}^{a}\cos\left[\frac{\pi }{\lambda z}{\left({x-x}_{1}\right)}^{2}\right]{\mathrm{dx}}_{1}+\\ & i\int_{0}^{a}\sin\left[\frac{\pi }{\lambda z}{\left({x-x}_{1}\right)}^{2}\right]{\mathrm{dx}}_{1} \end{array}$$

As aforementioned, the objective is to make the distribution of light intensity stable at any position within one Talbot distance. Optical linear encoders equipped with 20 μm pitch period gratings are the most commonly used in the industrial sector. Assuming the pitch lines are equal, then the value of a is 10 μm, as seen in Figure 3. The wavelength of the desired LED light source is 875 mm. Based on Equation (1), the Talbot distance is 914 µm under these conditions. From the perspective of practical applications, the distance between the reading head and main scale is usually greater than 400 μm to avoid friction or collisions, and 400 μm is approximately half of the Talbot distance in this optical system. Therefore, gap values are chosen as 400 μm, 800 μm, and 900 μm for theoretical calculation. The scanning reticle can be considered a continual series of light sources, and the distribution of light intensity on the main scale is the composition of all. Because the distribution of light intensity by each light source can be calculated according to Equation (6), the total distribution of light intensity is computable. The data from the scanning reticle made by amplitude grating are given in Figure 5; the phase is reversed at the distance of 400 μm when amplitude grating is applied. As the objective of adding phase grating is to introduce different phases between the light sources on the scanning reticle, the same method of calculation still applies. The trials on the calculation method conclude with the phase shifted by π/2 between adjacent light sources; the phase of the light distribution is the most stable with different distances, as illustrated in Figure 6; and the phase of light distribution is the same between 400, 800, and 900 μm mounting gaps.

Based on the theoretical calculation, the proposed pattern of the scanning reticle is illustrated in Figure 3b. The orange columns represent the phase grating, while the other parts are identical to the traditional scanning reticle in Figure 3a. The widths of white, gray, and orange columns are set to a, and with such an arrangement, the fabrication of the scanning reticle can be greatly simplified.

### 2.2. Simulations

An additional simulation was made using Virtual Lab Fusion to verify the abovementioned calculation results. Virtual Lab Fusion is a simulation platform based on physical optics; it is ideal for simulating complex light propagations. The mounting distance is set as 400, 800, and 900 μm between scanning reticles and the main scale. The results of the scanning reticles made by amplitude grating are demonstrated in Figure 7, Figure 8 and Figure 9, which show the phase of light intensity distribution is unstable at different mounting distances. Figure 7a, Figure 8a and Figure 9a exhibit the single-dimensional distribution of light intensity at the mounting distances of 400, 800, and 900 μm, respectively. The phase of light is identical between Figure 8a and Figure 9a; in contrast, there is a π/2 phase difference between the light distribution shown in Figure 7a. The dual dimensional distribution of light intensity is shown by Figure 7b, Figure 8b and Figure 9b. The bright and dark areas in Figure 8b and Figure 9b are identical, but Figure 7b shows the opposite pattern. Moreover, the unexpected light lines with a width of 4 μm appear at the region between bright and dark in Figure 8b, where the variation in phase may impede the formation of the Moire fringe. Based on these simulation results, the image of the diffraction light at 400 μm (close to half of one Talbot distance) is almost reversed compared with the results at 800 and 900 μm (close to one Talbot distance). Such a phenomenon resembles the previous calculation results. The instability of light intensity distribution and phase drift ton signals not only leads to incorrectness in positions but can also be difficult to read by photocells in the reading head. That is to say, no expected output electrical signals can be generated in this case. On the other hand, the phase remains stable, and the bright and dark areas are not swapped in the simulation results of the scanning reticle made by amplitude and phase grating for the distances of 400, 800, and 900 μm (as shown in Figure 10, Figure 11 and Figure 12). The simulations by Virtual Lab Fusion demonstrate that the amplitude and phase grating outperform their counterpart and have good potentiality to keep the distribution of light stable when mounting distance varies.

### 2.3. Fabrication

The fabrication of the proposed scanning reticle can be completed using two steps. The first step is to fabricate the amplitude grating, which is quite similar to the mainstream technology of making the optical linear encoder’s glass scales nowadays (Figure 13). Step one contains the processes of coating, photolithography, and etching. The chrome is coated onto glass substrates by e-beam evaporation; the uniformity is supposed to be finer than 5%. The source of UV irradiation is a 365 nm wavelength lamp with an exposure of 100 mJ. To make a desirable amplitude grating, all chrome must be etched away in designed areas until the glass substrate is exposed. Meanwhile, timing is critical to ensure the width and straightness of the chrome lines.

Step two (Figure 14) resembles the previous step. It is noteworthy that the exposure process can only take place after the mask and amplitude grating are perfectly aligned. The accuracy of alignment has a significant impact on the line width and period on the scanning reticle. In this experiment, it was performed under a microscope with an accuracy of ±1 μm for the alignment. After the photolithography process, a transparent film made of TiO_2_ is deposited onto the scanning reticle. Temperature is critical in this process to prevent the deformation of the photoresist. Finally, the photoresist and the TiO_2_ film on top of it are removed by the acetone solution. While the area of TiO_2_ film that has direct contact with the glass substrate is left. The remaining TiO_2_ film serves as the phase grating.

The scanning reticles made by amplitude and phase hybrid gratings were fabricated in accordance with the abovementioned steps in the laboratory. A photograph of the scanning reticle under an optical microscope is shown in Figure 15. The pitch period of the grating is 20 μm. The chrome-coated zones are the yellow strips in this photograph, while the red columns are the TiO_2_ film, and the black areas are the glass substrate as there is no reflective light. An alpha-step D-100 stylus profiler was employed to measure the height of the surface. The height of the chrome coating is about 200 nm, while the thickness of TiO_2_ film is 430 nm. By using such an arrangement, the phase of the incident light can be reversed by approximately π/2. The protuberant part in Figure 16 is the overlay of the TiO_2_ and chrome coating. The reason for this phenomenon is due to the limitation of laboratory production technology; the accuracy of the alignment in step two is ±1 μm. To avoid missing the TiO_2_ film between the designed grating lines, as illustrated in Figure 3b, the coating area of the TiO_2_ film was deliberately enlarged. Since the chrome zones are opaque, the overlay should have little impact on the distribution of light. The fabricated scanning reticle should be qualified to conduct experiments.

## 3. Investigation Methodology and Experiment Setup

The working principle of the open-type optical linear encoder is by using the photocells to collect the Moire fringe signals (light signals) generated by the motion between the scanning reticle and main scale and then convert them into the sine and cosine electrical output signals by the electronics systems. The period of the sine and cosine signals is usually equal to the grating pitch, which is usually at 20 μm or 40 μm, and must be interpolated for finer resolution in practical applications. The amplitude and sinusoidal quality are the key parameters of the output electrical signals for the interpolation process.

In the theoretical and simulation parts, the distributions of light intensity with different scanning reticles are analyzed. The scanning reticle with the combination of amplitude and phase grating has stable light patterns at different distances, which is promising for increasing the mounting tolerance with fine optical signals. Stable light signals are a prerequisite to generating high-quality output electrical signals. However, as the measured displacement positions are commonly interpreted by electrical signals in industrial applications, it is more convincing to directly experiment on the electrical signals of the optical linear encoders equipped with the proposed scanning reticles. Prototypes of open-type linear encoder’s reading heads were assembled to perform the experiment. A schematic of the reading head is illustrated in Figure 17.

As aforementioned, the priority of the experiment is to investigate the sinusoidal quality and variation in the electrical signal’s amplitude with different mounting distances and with different types of scanning reticles. The output electrical signal that can meet both key requirements at a mounting distance can be viewed as the optical linear encoder that can be fully functional at such a distance. The prototypes of the encoder’s reading heads need to be made to conduct the experiments. Three scanning reticles with a combination of amplitude and phase grating were fabricated in the laboratory based on two steps in Section 2.3, while two other scanning reticles with amplitude grating were produced following just the process of step one. These five scanning reticles were assembled into five prototypes of reading heads, while other key components, such as optical systems, photocells, and electronics system were identical. A reflective stainless-steel tape was fixed on the optics bench, which served as the main scale. The fabricated reading heads can be attached to the vertical stage, with a travel range of 5 mm and a sensitivity of 3 μm to alter different mounting distances. Fifteen different mounting distances were selected from 0.30 mm to 1.00 mm to investigate the variation in output electrical signals. An oscilloscope from Agilent Technologies was employed to capture and analyze the signals. In practical applications, optical linear encoders can be functional when the signals show the sinusoidal and amplitude ratio is 0.6 or higher. A schematic of the experiment is illustrated in Figure 18.

The experimental equipment consists of:Five prototypes of the optical linear encoder’s reading heads with two different types of scanning reticles inside.A Zolix TSMV5-1A vertical stage, with a travel range of 5 mm, and a sensitivity of 3 μm to control the mounting distance between the reading head and main scale.A reflective stainless-steel tape made by Precizika Meteorology UAB, and the pitch period of the incremental track is 20 μm.An oscilloscope from Agilent Technologies to observe the output electrical signals.

## 4. Experimental Results and Discussion

Five reading heads were fixed on the experimental bench to analyze the output electrical signals at each mounting distance. The mounting distance was adjustable from 0.30 mm to 1.00 mm with a step of 0.05 mm. The reading heads had a transverse relative motion of 0.1 m/s with the main scale so that the sine and cosine electrical output signals could be observed on the oscilloscope. At each mounting distance, the output electrical signals must generate a sinusoidal signal to be considered valid points. Reading heads 1–3 were the ones equipped with the scanning reticle made by the combination of amplitude and phase grating, while reading heads 4 and 5 contained the scanning reticle with amplitude grating. In the case where there were no sinusoidal signals captured, the selected distance was considered as not workable for the optical linear encoder and, thus, was not recorded in the diagram in Figure 19. The experimental results indicate that all reading heads 1–3 have sinusoidal signals from the distance between 0.30 mm and 1.00 mm. However, for reading heads 4 and 5, only at distances of 0.40, 0.45, 0.90, and 0.95 mm could the sinusoidal signals be observed. All these positions were recorded with the signal amplitude ratio as well. In practical applications, optical linear encoders can be functional when the signal amplitude ratio is 0.6 or higher. Based on the experiment results in Figure 19, reading heads 1–3 could be functional from a 0.45 to 1.00 mm mounting distance, while reading heads 4 and 5 were only functional at distances of 0.40, 0.45, 0.90, and 0.95 mm.

With the results of the experiment, the following conclusions can be given:The results using reading heads 4–5 indicate that the optical linear encoder with the traditional scanning reticle can only operate at distances of 0.40, 0.45, 0.90, and 0.95 mm. With such rigorous requirements, the mechanical surface for the installment must be perfectly leveled, and a high-precision mounting device must be employed. On the other hand, with the proposed scanning reticle, reading heads 1–3 indicate that the effective mounting distance can range from 0.45 to 1.00 mm. If the instructed mounting distance of the optical linear is set as 0.72 mm, then the mounting tolerance is ±0.275 mm. Such mounting tolerance makes the installation of the optical linear encoder without a high-precision mounting device possible and can reduce errors incurred by incorrect mounting distance.The reading heads 1–3 show a similar workable mounting distance range and trend. Nonetheless, there are still deviations at some mounting distances. For instance, at 0.40 mm, the signal amplitude ratio is 0.83, 0.53, and 0.59. This is possibly caused by the slight angle difference in the scanning reticles when fixed onto the reading heads since there was no angle measurement equipment for this experiment. The theoretical model of the light distribution with the angle difference between the scanning reticle and main grating should be developed based on generalized grating imaging for future work. The angle measurement equipment should be built for more precise experiments and investigations.The theoretical calculations and Virtual Lab simulations indicate that the light distribution intensities are stable with the proposed scanning reticle, which means that the quality of optical signals should be adequate for the subsequent electronic system to generate sine and cosine waves. As for the traditional scanning reticles, the light distribution intensities were unstable, which suggests that the quality of optical signals is unlikely to be readable by the electronics system during the entire mounting range. The Experimental Results support such assumptions.Other characteristics of the optical linear encoders can be improved in the future based on this work. For reading heads 1–3, the signal amplitude ratios were all above 0.6 for the entire range from 0.45 to 1.00 mm, but the deficiency was that they are not uniform, and this could lead to a different quality of the interpolation of the electrical signals. Future work plans will focus on developing an algorithm to compensate for the electrical signals.

## 5. Conclusions

For open-type optical linear encoders, the correctness of mounting distance is critical for accuracy. However, in industrial applications, it is almost impossible to install all reading heads and main scales at perfect positions and keep the distance extremely stable during the entire measurement range. The optimum solution for this problem is to increase the mounting tolerance. This research proposed combined amplitude and phase grating as the scanning reticle for improvement. The theoretical calculation and Virtual Lab simulation results indicate that the stability of light distribution using the proposed scanning reticle outperforms its counterpart in mounting distances between 0.45 and 1.00 mm. Finally, experiments were performed to verify the theoretical results. A prototype of reading heads equipped with the proposed scanning reticle was fabricated in the laboratory. The experimental results indicate that the output electrical signals from the reading heads equipped with the proposed scanning reticle were sufficient for open-type optical linear encoders to be functional within a mounting range from 0.45 to 1.00 mm. This work lays the foundations for future research to develop high-accuracy open-type optical linear encoders with fine mounting tolerance.

## Figures and Tables

**Figure 1 sensors-23-01987-f001:**
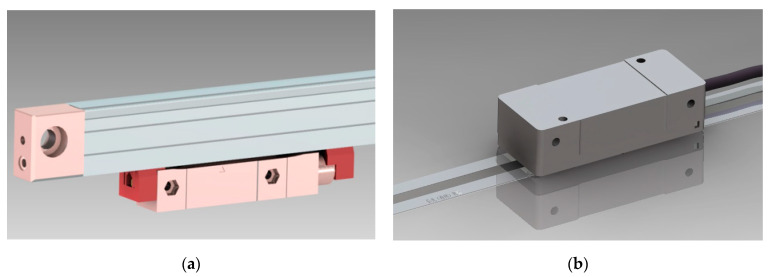
Optical linear encoder in different mechanical structures. (**a**) Enclosed-type optical linear encoder. (**b**) Open-type linear encoder.

**Figure 2 sensors-23-01987-f002:**
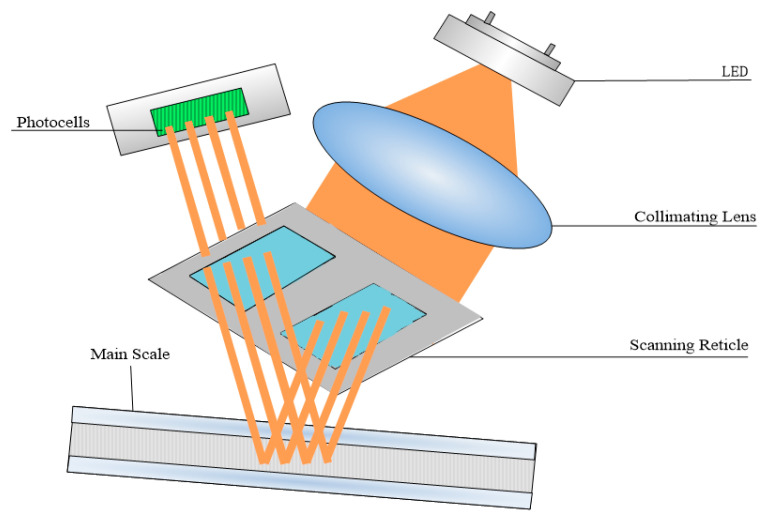
Optical path of open-type linear encoder.

**Figure 3 sensors-23-01987-f003:**
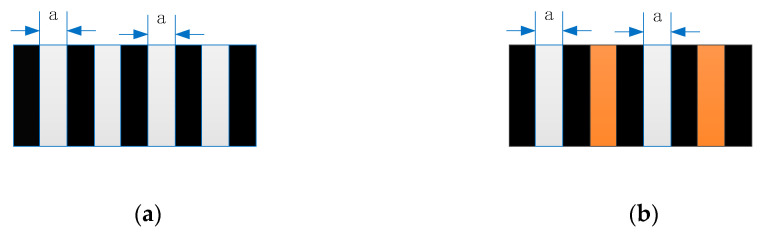
Different types of scanning reticles. (**a**) A traditional scanning reticle made by amplitude grating. (**b**) The proposed scanning reticle made by combination of amplitude and phase grating.

**Figure 4 sensors-23-01987-f004:**
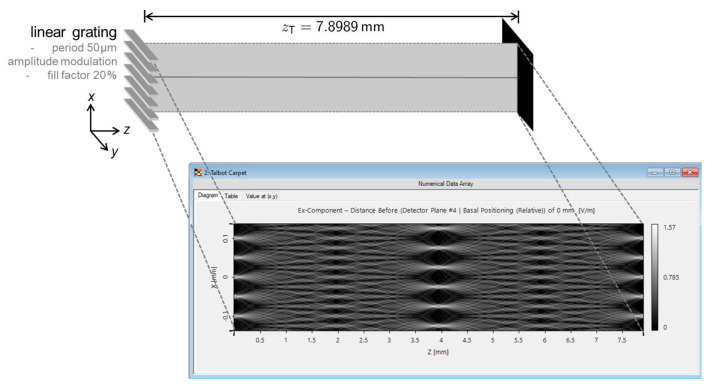
Light distribution of the Talbot effect (www.lighttrans.com, accessed on 6 February 2023).

**Figure 5 sensors-23-01987-f005:**
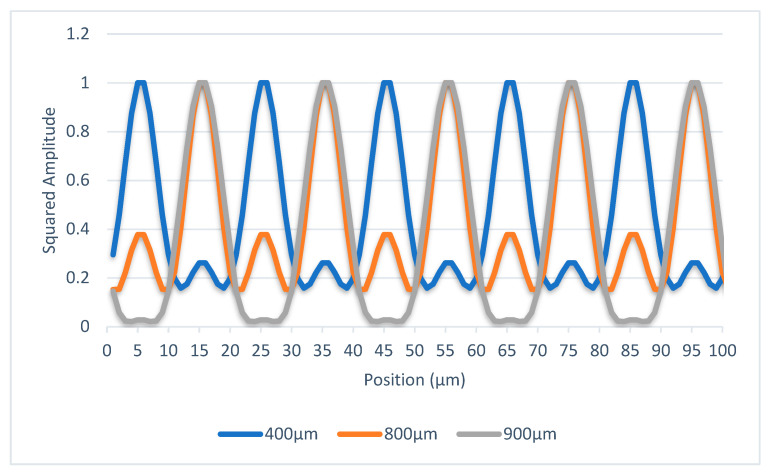
Distribution of light intensity with different scanning reticles at three mounting distances through the scanning reticle made by amplitude grating.

**Figure 6 sensors-23-01987-f006:**
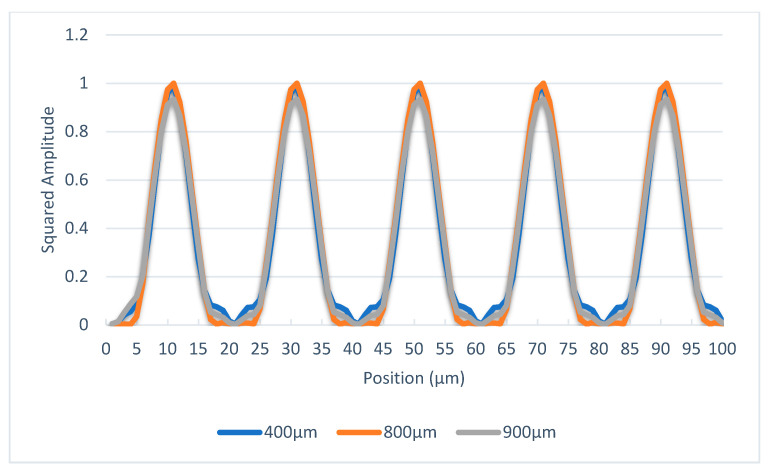
Distribution of light intensity with different scanning reticles at three mounting distances through the scanning reticle made by combination of amplitude and phase grating.

**Figure 7 sensors-23-01987-f007:**
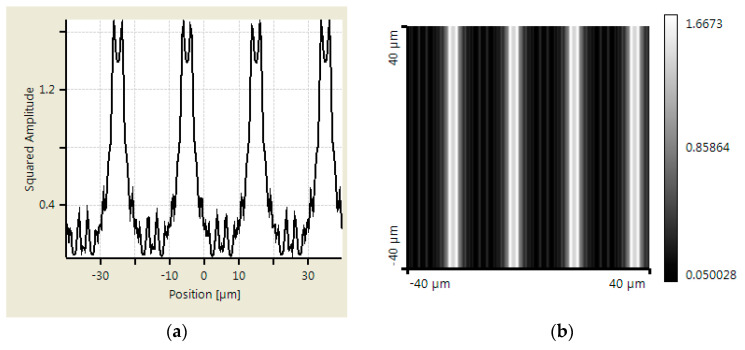
Distribution of light intensity at 400 μm with scanning reticle made by amplitude grating. (**a**) Single dimensional. (**b**) Dual dimensional.

**Figure 8 sensors-23-01987-f008:**
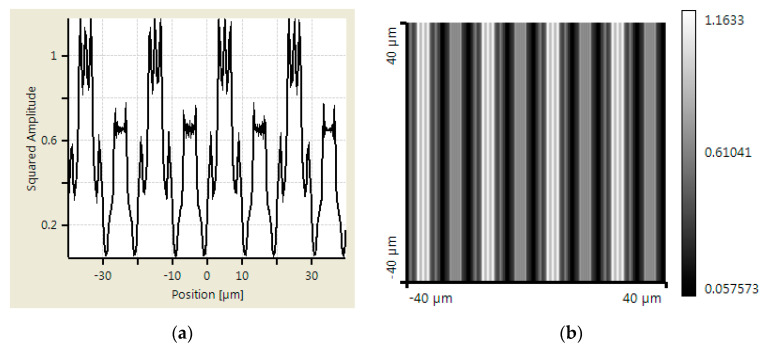
Distribution of light intensity at 800 μm with scanning reticle made by amplitude grating. (**a**) Single dimensional. (**b**) Dual dimensional.

**Figure 9 sensors-23-01987-f009:**
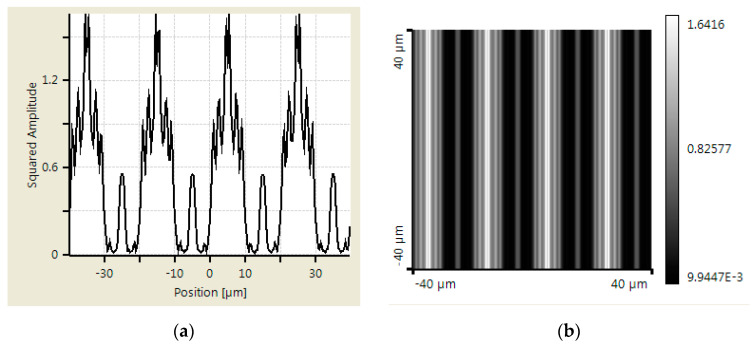
Distribution of light intensity at 900 μm with scanning reticle made by amplitude grating. (**a**) Single dimensional. (**b**) Dual dimensional.

**Figure 10 sensors-23-01987-f010:**
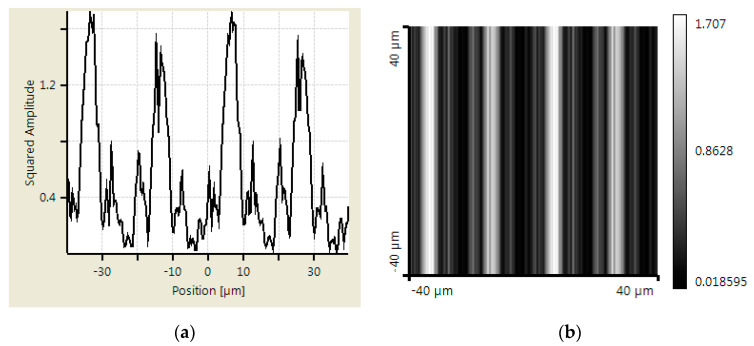
Distribution of light intensity at 400 μm with scanning reticle made by combination of amplitude and phase grating. (**a**) Single dimensional. (**b**) Dual dimensional.

**Figure 11 sensors-23-01987-f011:**
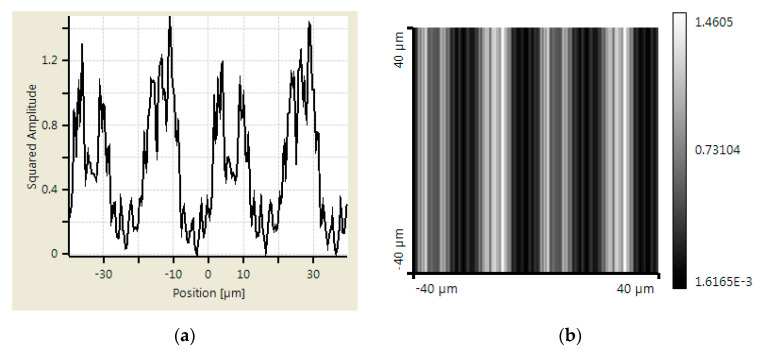
Distribution of light intensity at 800 μm with scanning reticle made by combination of amplitude and phase grating. (**a**) Single dimensional. (**b**) Dual dimensional.

**Figure 12 sensors-23-01987-f012:**
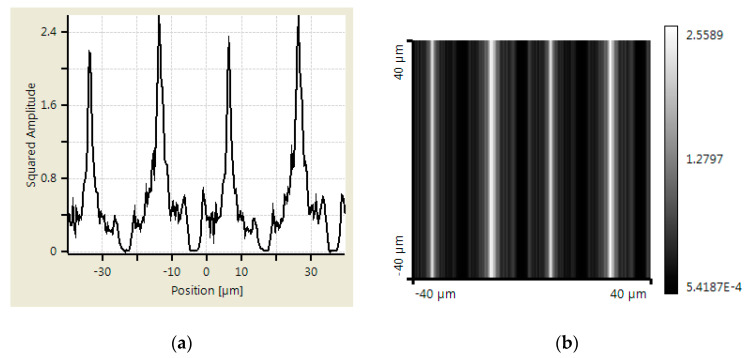
Distribution of light intensity at 900 μm with scanning reticle made by combination of amplitude and phase grating. (**a**) Single dimensional. (**b**) Dual dimensional.

**Figure 13 sensors-23-01987-f013:**
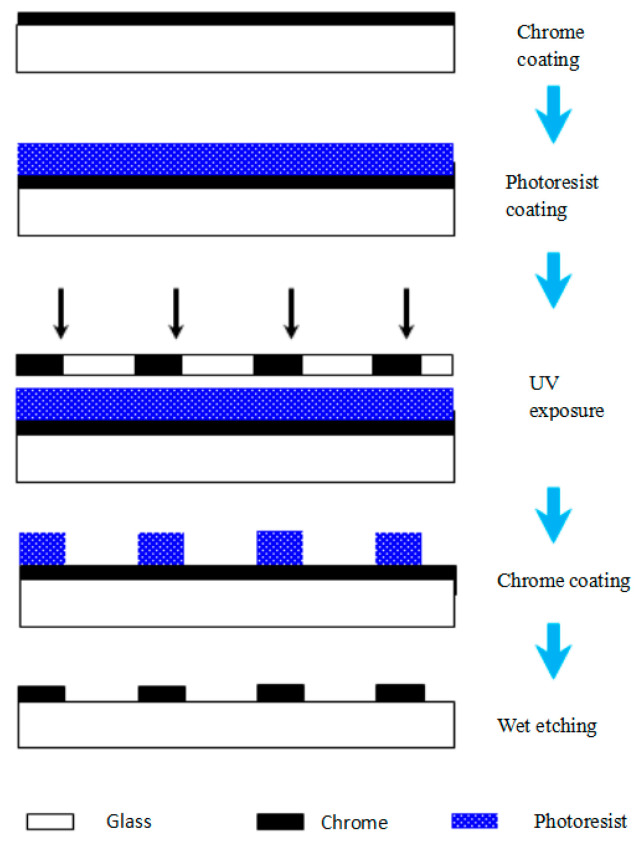
Step one of fabrication.

**Figure 14 sensors-23-01987-f014:**
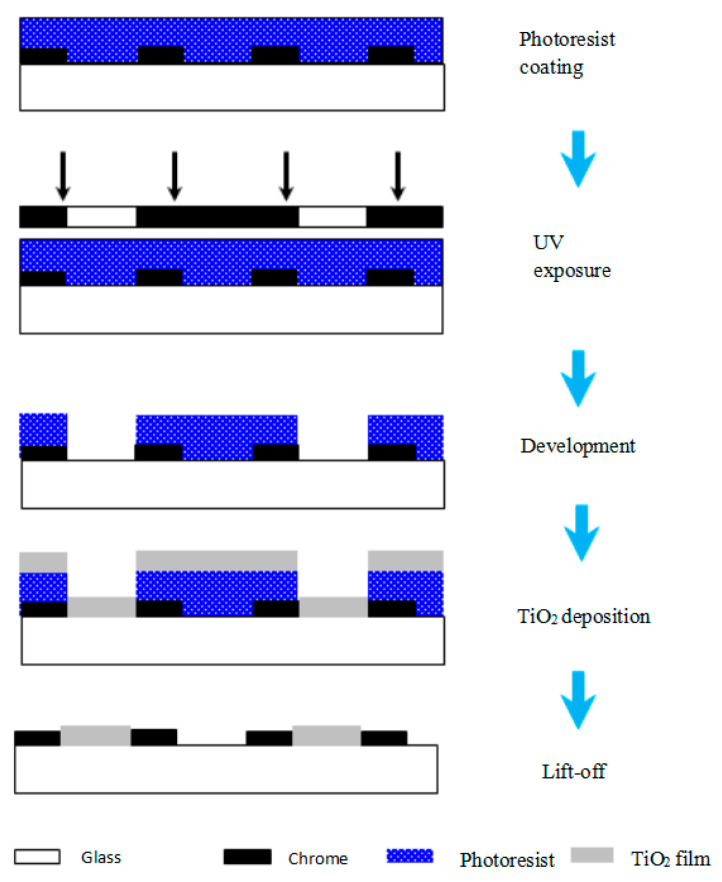
Step two of fabrication.

**Figure 15 sensors-23-01987-f015:**
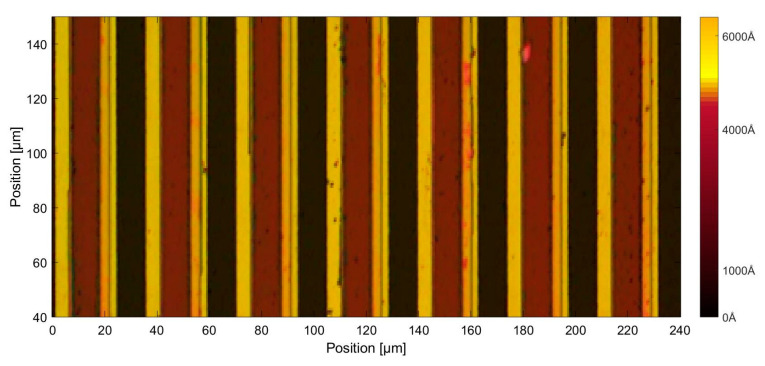
Proposed scanning reticle under an optical microscope.

**Figure 16 sensors-23-01987-f016:**
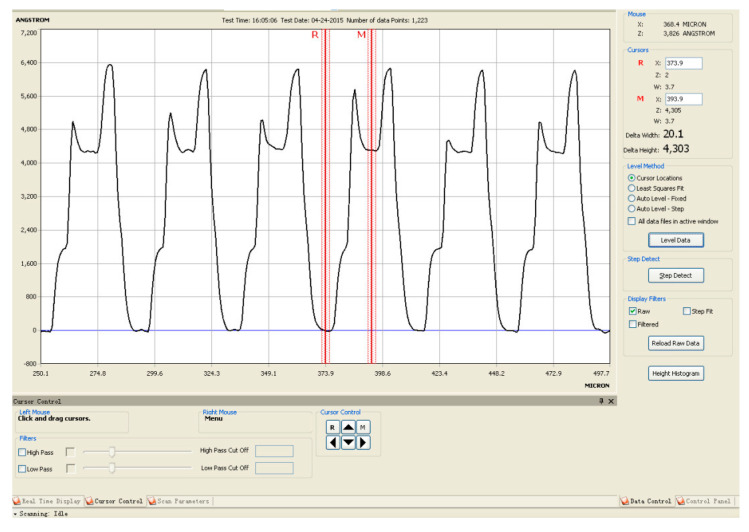
Proposed scanning reticle under an alpha-step D-100 stylus profiler.

**Figure 17 sensors-23-01987-f017:**
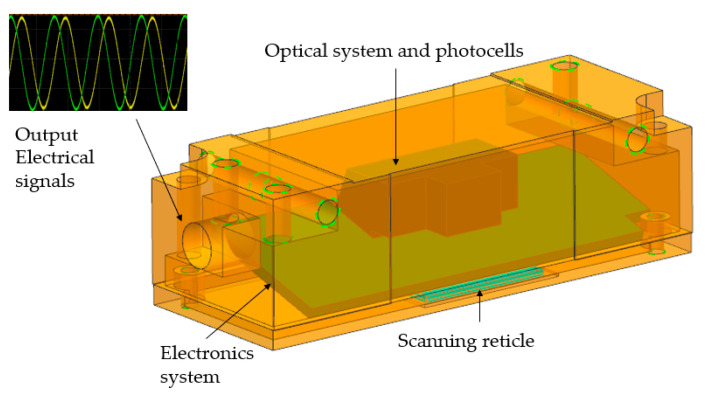
Schematic of the reading head of the open-type optical linear encoder.

**Figure 18 sensors-23-01987-f018:**
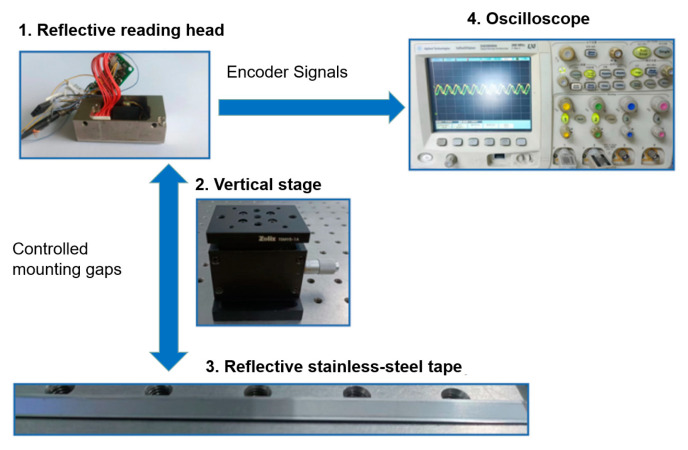
Schematic view of experiment setup.

**Figure 19 sensors-23-01987-f019:**
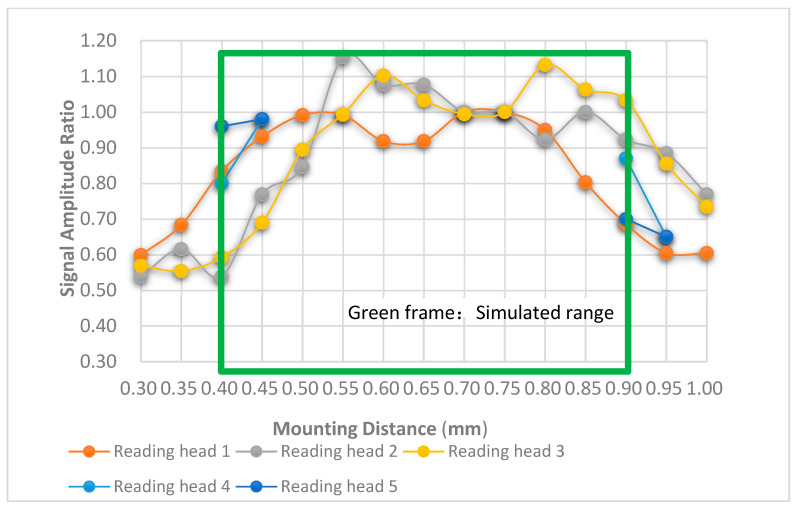
Signal amplitude at different mounting distances.

## Data Availability

Not applicable.

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
