# Peer review of "A Method to Improve Mounting Tolerance of Open-Type Optical Linear Encoder"

_sensors, 2023, doi:10.3390/s23041987_

Round 1

Reviewer 1 Report

Dear Authors

Thank you for submitting this interesting and relevant study.

The results are innovative and the methodology is sound.

The conclusions are supported by the results.

The paper is overall correctly written and clearly structured, but some improvements are necessary.

The submission could be considered for publication provided that the Authors address the following comments.

General comments

- Always have space between the measure and the unit (e.g., it is '40 µm' and not '40µm').

- In many instances the terms 'accuracy' and 'precision' are not correctly used; according to VIM (International Vocabulary of Metrology),'accuracy' refers to closeness of a measurement to the target or true (e.g., calibration) value, while 'precision' refers to the repeatability of several measurements executed under the same operating conditions. Often, when the Authors write 'precisely', they in fact mean 'accurately'. Please correct accordingly.

- The Authors are recommended to extend section 3 with additional experiments and results.

Detailed comments

- line 46 = change '...source of vibration.[15]. While...' to '...source of vibration [15]. While...'

- line 97 = change '...theory of Talbot effect[19],...' to  '...theory of Talbot effect [19],...' 

- line 106 = change '...LightTrans demonstrated[20],...' to  '...LightTrans demonstrated [20],...' 

- line 124 = change '...distribution of light intensity[21].Nevertheless, such...' to '...distribution of light intensity [21]. Nevertheless, such...'

- line 129 = change '...theory. In Fig.3, each...' to '...theory. In Fig. 3, each...'

- Figure 5 = the axis X is awkward: it should start at 0 µm and have intervals at 5 µm increment (e.g., 0, 5, 10, 15,...); axis X caption should be: Position [µm] (without the comma)

- Figure 6 = the axis X is awkward: it should start at 0 µm and have intervals at 5 µm increment (e.g., 0, 5, 10, 15,...); axis X caption should be: Position [µm] (without the comma)

- line 171 = change '...by Fig .3(b). The orange...' to '...by Fig. 3(b). The orange...'

- line 178-179 = change '...Fig 178 .7, Fig8 and Fig. 9, which...' to '...Fig. 7, Fig. 8 and Fig. 9, which...'

- line 190 = change '...(a) Single dimensional;(b) Dual dimensional.' to '...(a) Single dimensional; (b) Dual dimensional.'

- line 192 = change '...(a) Single dimensional;(b) Dual dimensional.' to '...(a) Single dimensional; (b) Dual dimensional.'

- line 194 = change '...(a) Single dimensional;(b) Dual dimensional.' to '...(a) Single dimensional; (b) Dual dimensional.'

- line 196 = change '...(a) Single dimensional;(b) Dual dimensional.' to '...(a) Single dimensional; (b) Dual dimensional.'

- line 198 = change '...(a) Single dimensional;(b) Dual dimensional.' to '...(a) Single dimensional; (b) Dual dimensional.'

- line 200 = change '...(a) Single dimensional;(b) Dual dimensional.' to '...(a) Single dimensional; (b) Dual dimensional.'

- Figure 15 = please include an horizontal scale bar in the image for the X direction and a color bar for the Z heights

- line 233 and 234 = the sentence 'The reason for this phenomenon is due to the limitation of laboratory production technology, it’s almost impossible to seamless align the TiO2 film perfectly with chrome zones,...' should be replaced by a clear statement of the limits in terms of repeatability and accuracy of the alignment; qualitative statements such as 'almost impossible' should be avoided

- line 268 = change 'Three proposed scanning reticle was fabricated...' to 'Three proposed scanning reticles were fabricated...'

- line 277-278 = the sentence 'The experimental results are highly coinciding with the outcome of calculations and simulations.' should be replaced by a clear statement of the repeatability and deviations between experimental and simulation results; qualitative statements of this kind should be avoided

- Figure 18 = in the X axis the scale should indicate 0.30, 0.35, 0.40,... and caption without comma; in the Y axis there is a spelling error in 'Signal'

- Figure 18 = please add also the results from simulation in the chart

Author Response

Dear reviewer, 

Thank you so much for your kind and insightful comments.

On behalf of all authors, I have revised the article accordingly.

Please see the attachment for the point-by-point response.

Best regards and thank you  

Xinji Lu

Reviewer 2 Report

-Figure 3 should be presented and explained in detail to improve understanding. For example, the white area in the background should be explained and possibly change color.

-From Equation 1, a picture showing the variables should be included. Figures 7 through 12 should also be explained in greater detail.

-The experimental or research methods should be thoroughly and clearly explained.

-Clearly explain the procedure, and add experimental results to compare with what is available in the previous product.

-Experimental results should be added and explained in more detail, comparing them to what is available in the previous product.

Conclusion: The results have been edited and included in the summary.

Comments and Suggestions for Authors:

There should be future recommendations that will be applied, as well as applications

Author Response

(The authors gave the same response as above.)
